# When to Memorize and When to Stop:
# Gated Recurrent Memory for Long-Context Reasoning

**Leheng Sheng** [1]  **Yongtao Zhang** [2]  **Wenchang Ma** [2]  **Yaorui Shi** [3]  **Ting Huang** [2]
**Xiang Wang** [3]  **An Zhang** [3]  **Ke Shen** [2]  **Tat-Seng Chua** [1]

## Abstract

While reasoning over long context is crucial for various real-world applications, it remains challenging for large language models (LLMs) as they suffer from performance degradation as the context length grows. Recent work MemAgent has tried to tackle this by processing context chunk-by-chunk in an RNN-like loop and updating a textual memory for final answering. However, this naive recurrent memory update faces two crucial drawbacks: (i) memory can quickly explode because it can update indiscriminately, even on evidence-free chunks; and (ii) the loop lacks an exit mechanism, leading to unnecessary computation after even sufficient evidence is collected. To address these issues, we propose GRU-Mem, which incorporates two text-controlled gates for more stable and efficient long-context reasoning. Specifically, in GRU-Mem, the memory only updates when the update gate is open and the recurrent loop will exit immediately once the exit gate is open. To endow the model with such capabilities, we introduce two reward signals $r^{\text{update}}$ and $r^{\text{exit}}$ within end-to-end RL, rewarding the correct updating and exiting behaviors respectively.

## 1. Introduction

Reasoning over extremely long contexts is a crucial capability of large language models (LLMs) for real-world applications (OpenAI, 2023; DeepSeek-AI et al., 2025; Wan et al., 2025), such as reading an entire book or processing large-scale memories in agentic systems (Packer et al., 2023; Li et al., 2025; Chhikara et al., 2025). While impor-

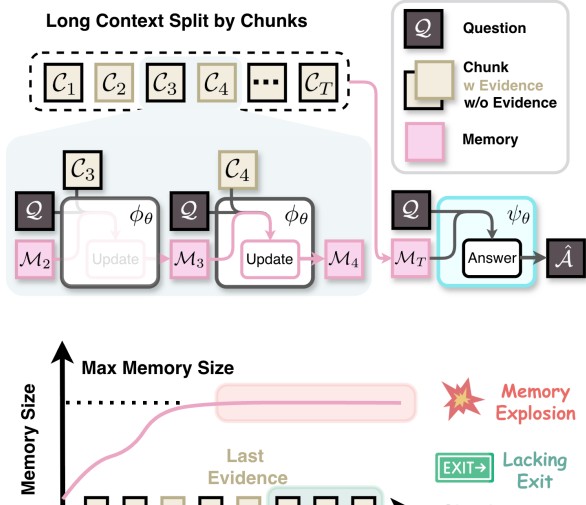

*Figure 1.* MemAgent and its limitations. The MemAgent reads a long context chunk-by-chunk in an RNN-like manner, recurrently updating a textual memory and answering from the final memory. It faces two crucial risks: memory explosion by over-accumulating irrelevant memories and lacking an exit mechanism when collected sufficient evidences.

tant, this capability remains challenging: LLMs typically experience dramatic performance degradation as the context length grows (Hsieh et al., 2024; Bai et al., 2024), and they struggle to handle corpora that exceed their maximum context window (Yu et al., 2025a).

To address these limitations, recent work MemAgent (Yu et al., 2025a) has explored a recurrent memory paradigm for long context reasoning in an RNN-like manner (Yu et al., 2025a; Li et al., 2025; Mienye et al., 2024). As illustrated in Figure 1, instead of encoding the entire context in a single forward pass (Su et al., 2024a; Xiong et al., 2024), this method reformulates long-context reasoning as a sequential, recurrent, and chunk-by-chunk memorization process. Concretely, the entire long context $\mathcal{C}$ is divided into $T$ fixed-size chunks $\{\mathcal{C}_1 \cdots \mathcal{C}_T\}$ first. At step $t$, a memory agent $\phi_\theta$ reads the question $\mathcal{Q}$, chunk $\mathcal{C}_t$, and the previous memory $\mathcal{M}_{t-1}$ to recurrently update a textual memory $\mathcal{M}_t$. After processing all chunks, an answer agent $\psi_\theta$ predicts the answer $\hat{\mathcal{A}}$ conditioned on the final memory $\mathcal{M}_T$ and

[1]National University of Singapore [2]Seed, ByteDance [3]University of Science and Technology of China. Correspondence to: An Zhang <an_zhang@ustc.edu.cn>, Ke Shen <shenke@bytedance.com>.

*Proceedings of the 43rd International Conference on Machine Learning*, Seoul, South Korea. PMLR 306, 2026. Copyright 2026 by the author(s).

the question $\mathcal{Q}$. In practice, the memory agent $\phi_\theta$ and the answer agent $\psi_\theta$ share the same parameterized policy model $\theta$, with their behaviors differentiated by the prompt assignment. Trained end-to-end with reinforcement learning (RL) (Yu et al., 2025a;b), this recurrent memory paradigm shows potential in mitigating the performance degradation from ingesting an overly long context at once, and ideally breaks the context window size limit.

However, the vanilla approach inherits several drawbacks of naive RNN-style memory updates (Bengio et al., 1994; Hochreiter & Schmidhuber, 1997). Here, we identify two key limitations that may hinder the memory stability and efficiency of reasoning in practice:

- **Risk of Memory Explosion.** As shown in Figure 1, when updating on evidence-free chunks, the memory agent may accumulate irrelevant or noisy content over time. This drift progressively inflates the memory, potentially exceeding the allotted budget and triggering *memory explosion*. Once already exploded, the accumulated noise can further impede subsequent updates, making it harder to incorporate new key evidence from later chunks (Yu et al., 2025a). Additionally, inefficiently regenerating the already-exploded memory at each step also increases the inference cost. As a result, such risks of memory explosion undermine both long-horizon stability and runtime efficiency for long-term reasoning.

- **Lack of Exit Mechanism.** As shown in Figure 1, the vanilla workflow is hard-coded to process all chunks and provides no early-exit mechanism when the collected evidence is sufficient. That is, even after sufficient evidence has been collected already (*e.g.,* the last necessary evidence for the answer has appeared), the model must still process the remaining chunks until the end, incurring avoidable computation. This inefficiency is amplified when evidence is unevenly distributed (*e.g.,* when reranking brings key evidence early (Fan et al., 2024; Zhang et al., 2025b)) This inefficiency becomes even more severe when evidence is unevenly distributed across the context, where in some cases key evidence may appear very early (*e.g.,* when the long context is reordered by specific reranking techniques) (Fan et al., 2024; Zhang et al., 2025b).

To this end, we propose GRU-Mem, a gated recurrent memory framework for stable and efficient long-context reasoning, inspired by the effectiveness of gating in GRUs (Cho et al., 2014) for addressing long-term sequence modeling challenges in RNNs (*e.g.,* gradient explosion (Hochreiter & Schmidhuber, 1997)). The key idea is to augment the recurrent workflow with two text-controlled gates — an update gate (UG) and an exit gate (EG) — which decide (i) whether the memory should be updated on the current chunk and (ii) whether the model can terminate early once sufficient evidence has been collected. Specifically, at step $t$, the memory agent $\phi_\theta$ generates three key information: the update gate status $\mathcal{U}_t$, the candidate memory $\hat{\mathcal{M}}_t$, and the exit gate status $\mathcal{E}_t$, which can be formulated as $\mathcal{U}_t, \hat{\mathcal{M}}_t, \mathcal{E}_t = \phi_\theta(\mathcal{Q}, \mathcal{C}_t, \mathcal{M}_{t-1})$. Only when the update gate status $\hat{\mathcal{U}}_t$ is True, the memory $\mathcal{M}_t$ will be updated with the the candidate memory $\hat{\mathcal{M}}_t$, otherwise adopting the previous memory $\mathcal{M}_{t-1}$. Additionally, when the exit gate status $\mathcal{E}_t$ is True, indicating the last evidence required occurs in this chunk, the workflow can end immediately for answering the question. Finally, similar to the vanilla design, the answer agent $\psi_\theta$ provides the answer $\hat{A}$ based on the terminal memory $\mathcal{M}_t$. To endow the model with such capabilities, we introduce two reward signals $r^{\text{update}}$ and $r^{\text{exit}}$ within end-to-end RL, rewarding the correct updating and exiting behaviors, respectively. The update gate enables selective memory updates on only a few informative chunks, mitigating memory explosion, while the exit gate provides a flexible early-termination mechanism that reduces unnecessary computation; together, they yield more stable and efficient long-context reasoning.

We further conduct extensive experiments to verify the effectiveness of GRU-Mem. First, we show that GRU-Mem generally outperforms the vanilla MemAgent across diverse tasks and among different model sizes, and generally achieves up to 400% times inference speed acceleration (*cf.* Section 4.1). Second, we validate the effectiveness of the two introduced gating mechanisms, where GRU-Mem can reduce the risks of memory explosion and provide a meaningful exit mechanism (*cf.* Section 4.2). Third, we systematically analyze the role of components in GRU-Mem with the ablation study, including the training dynamics (*cf.* Section 4.3). We highlight that GRU-Mem provides a stable and efficient long context reasoning paradigm with the introduction of two gating mechanisms.

## 2. Preliminary

We briefly introduce the task formulation of reasoning over long contexts in Section 2.1 first. After that, we present the recurrent memory paradigm, as introduced in previous work MemAgent (Yu et al., 2025a), in Section 2.2.

### 2.1. Task Formulation

**Long-context QA**. In this paper, we primarily focus on the widely adopted long-context question answering (QA) tasks (Yu et al., 2025a; Hsieh et al., 2024). In these tasks, each example consists of a question $\mathcal{Q}$, a long context $\mathcal{C}$, and a ground-truth answer $\mathcal{A}$, which can be denoted as a triplet $(\mathcal{Q}, \mathcal{C}, \mathcal{A})$. Here, the context length $|\mathcal{C}_i|$ can be very large, such as millions of tokens.

**Evidence Sparsity.** To answer the question $\mathcal{A}_i$, LLMs are required to accurately locate and then reason over a few evidence pieces $\{e_k\}_{k=1}^K$, which are sparsely located within the long context $\mathcal{C}_i$. The sparse evidence distribution in long context reasoning is also commonly known as the needle in a haystack (NIAH) (Kamradt, 2023) problem.

## 2.2. Recurrent Memory for Long-context Reasoning

**Context chunks.** In the recurrent memory for long-context reasoning paradigm, the long context is split as a set of fixed-size context chunks $\mathcal{C} = \{\mathcal{C}_1 \cdots \mathcal{C}_T\}$ first. After splitting into chunks, the evidence is sparsely distributed in only a few chunks. That is, most of the chunks do not contain any information for answering the question $\mathcal{Q}$. This splitting strategy avoids directly feeding the entire context into LLMs at once.

**Workflow.** The recurrent memory paradigm for long-context reasoning adopts an RNN-like workflow with two agents: a memory agent $\phi_\theta$ for recurrently updating a textual memory and an answer agent $\psi_\theta$ for answering the question based on the final memory. Specifically, as shown in Figure 1, at each step $t$ the memory agent $\phi_\theta$ takes the question $\mathcal{Q}$, the current chunk $\mathcal{C}_t$, and the previous memory $\mathcal{M}_{t-1}$ as input, and generate an updated memory $\mathcal{M}_t$ in a recurrent manner, which can be formulated as:

$$\mathcal{M}_t = \phi_\theta(\mathcal{Q}, \mathcal{C}_t, \mathcal{M}_{t-1}). \tag{1}$$

After reading all the $T$ chunks, an answer agent $\psi_\theta$ generate the answer $\hat{\mathcal{A}}$ to the question $\mathcal{Q}$ based on the final memory $\mathcal{M}_T$, as formulated as follows:

$$\hat{\mathcal{A}} = \psi_\theta(\mathcal{Q}, \mathcal{M}_T). \tag{2}$$

Here, the memory agent $\phi_\theta$ and the answer agent $\psi_\theta$ are the same parameterized policy model $\theta$ differentiated by the prompt assignment.

**Workflow optimization with end-to-end RL.** This agent workflow can be optimized with the Multi-Conv DAPO algorithm as proposed in MemAgent (Yu et al., 2025a), which is an extension of the group relative policy optimization (GRPO) algorithm (Shao et al., 2024) in the multi-turn scenario. The key idea lies in treating each conversation (*i.e.,* one individual memory or answer turn) as an independent optimization target, and then optimizing these independent targets with their corresponding advantages. Specifically, one workflow trajectory in group $g$ can be expressed as $(o_{g,1}, o_{g,2}, ..., o_{g,T_g})$, where $T_g$ denotes the total turn number of generated conversations. Here $o_{g,T_g}$ denotes the answer agent turn while $(o_{g,1}, o_{g,2}, ..., o_{g,T_g-1})$ denote the memory agent turns. Each conversation output $o_{g,t}$ at step $t$ consists of a sequence of tokens $(o_{g,t,1}, o_{g,t,2}, ..., o_{g,t,|o_{g,t}|})$, where $|o_{g,t}|$ is the number of output tokens in this conversa-

tion. Then the overall loss can be formulated as follows:

$$\mathcal{J}(\theta) = \mathbb{E}_{(\mathcal{Q},\mathcal{A})\sim\mathcal{D}, \{o_{g,t}\}_{g=1}^G \sim \pi_{\theta_{\text{old}}}(\cdot|Q, o_{g,t-1})}$$
$$\left[ \frac{1}{Z} \sum_{g=1}^G \sum_{t=1}^{T_g} \sum_{i=1}^{|o_{g,t}|} \left( \ell_{g,t,i}^{\text{clip}} - \beta\, D_{\text{KL}}(\pi_\theta \| \pi_{\text{ref}}) \right) \right],$$

where $\quad \ell_{g,t,i}^{\text{clip}} = \min\left( \rho_{g,t,i}(\theta)\hat{A}_{g,t,i}, \right.$

$$\left. \text{clip}(\rho_{g,t,i}(\theta),\, 1 - \varepsilon_{\text{low}},\, 1 + \varepsilon_{\text{high}})\hat{A}_{g,t,i} \right),$$

and $\quad Z = \sum_{g=1}^G \sum_{t=1}^{T_g} |o_{g,t}|.$

$$\tag{3}$$

Here $\pi_\theta$ and $\pi_{\text{ref}}$ denote the policy model and reference model, $\varepsilon_{\text{low}}$ and $\varepsilon_{\text{high}}$ denote the lower and higher clipping factors as introduced in DAPO (Yu et al., 2025b), and $\rho_{g,t,i}(\theta)$ refers to the importance sampling weight:

$$\rho_{g,t,i}(\theta) = \frac{\pi_\theta(o_{g,t,i} \mid \mathcal{Q}, o_{g,t,<i})}{\pi_{\theta_{\text{old}}}(o_{g,t,i} \mid \mathcal{Q}, o_{g,t,<i})}. \tag{4}$$

In this vanilla approach, all the conversations within one group $g$ share the same advantage, and the advantage without the normalization of standard deviation is calculated as (Liu et al., 2025c):

$$\hat{A}_{g,t,i} = r_g^{\text{outcome}} - \text{mean}(\{r_g^{\text{outcome}}\}_{g=1}^G). \tag{5}$$

The reward for the final answer correctness is calculated as:

$$r^{\text{outcome}} = \mathbb{I}(\texttt{is\_equiv}(\mathcal{A}, \hat{\mathcal{A}})), \tag{6}$$

where $\hat{\mathcal{A}}$ is the predicted answer, and $\mathbb{I}(\cdot)$ denotes the indicator function. This assigns a reward of $1$ if the answer $\hat{\mathcal{A}}$ produced by the answer agent $\psi_\theta$ matches the ground truth answer $\mathcal{A}$, and $0$ otherwise.

With such end-to-end RL training, the long-context reasoning capability of this agent workflow is substantially enhanced, even enabling small-sized models to outperform big models that ingest the entire context in a single pass. However, indiscriminately updating on all chunks and the lack of an early exit mechanism still largely limit the memory stability and efficiency of the vanilla approach.

## 3. Methodology

In this section, we introduce the Gated Recurrent Memory (GRU-Mem), a recurrent memory workflow equipped with two gating mechanisms, for stable and efficient long-context reasoning. We first introduce the workflow of GRU-Mem in Section 3.1. After that, we then detail how to train the workflow with GRU-Mem using end-to-end reinforcement learning, including the tailored reward design and advantage calculation, in Section 3.2. Finally, we briefly introduce the inference strategies in Section 3.3.

```
You should reason about whether the new
section contains useful information,
what to update, and what to do next
first between <think> and </think>.
If the new section contains useful
information about the problem, you
should first generate
<check>yes</check>.  After that, update
the new memory between <update> and
</update>.
If the new section does not contain
useful information about the problem,
you should first generate
<check>no</check>.  After that, you
should keep the previous memory
unchanged between <update> and
</update>.
In the end, if you haven't collected
enough information for the problem,
return <next>continue</next>.  ONLY
when enough information is collected,
return <next>end</next>.
```

*Figure 2.* Prompt of GRU-Mem (partial).

### 3.1. Gated Recurrent Memory Workflow

To address the risk of memory explosion and lack of an exit mechanism, we introduce two gating mechanisms in the recurrent workflow, the update gate (UG) and the exit gate (EG), inspired by the success of gating in GRUs (Cho et al., 2014) for addressing long-term sequence modeling.

Specifically, GRU-Mem mainly changes the memory agent $\phi_\theta$ while keeping the answer agent $\psi_\theta$ unchanged, extending the vanilla $\phi_\theta$ with two extra actions for gate control (Yu et al., 2025a). It outputs a candidate memory $\hat{\mathcal{M}}_t$ and two binary gating signals, $\mathcal{U}_t$ and $\mathcal{E}_t$, indicating whether to update the memory and whether to terminate the loop, respectively. This process is formulated as follows:

$$\mathcal{U}_t, \hat{\mathcal{M}}_t, \mathcal{E}_t = \phi_\theta(\mathcal{Q}, \mathcal{C}_t, \mathcal{M}_{t-1}). \tag{7}$$

To achieve this, the memory agent $\phi_\theta$ responds by following a structured output format (see prompts in Figure 2 and Appendix C, with case studies in Appendix E). The process of one single step $t$ of $\phi_\theta$ in GRU-Mem is illustrated in Figure 3. It first produces intermediate reasoning enclosed by `<think>` and `</think>`, and then emits an update decision within `<check>` and `</check>`, where `"yes"` (*i.e.,* $\mathcal{U}_t$ == True) triggers a memory update and `"no"` (*i.e.,* $\mathcal{U}_t$ == False) skips it. Next, it outputs the candidate memory $\hat{\mathcal{M}}_t$ within `<update>` and `</update>`. If $\mathcal{U}_t$ == True, the memory will be updated with the candidate memory (*i.e.,* $\mathcal{M}_t \leftarrow \hat{\mathcal{M}}_t$). If $\mathcal{U}_t$ == False, the previous memory $\mathcal{M}_{t-1}$ will be adopted while the candidate memory $\hat{\mathcal{M}}_t$ will be discarded (*i.e.,* $\mathcal{M}_t \leftarrow \mathcal{M}_{t-1}$). Finally, it decides whether to continue collecting evidence between `<next>` and `</next>`, where `"continue"` means continuing the recurrent loop and `"end"` means terminating

---

**Algorithm 1:** Long Context Reasoning with GRU-Mem

**Input:** Question $\mathcal{Q}$, the chunk size $s$, the whole context $\mathcal{C}$, and use_exit_gate: bool.
**Output:** Answer to the question $\hat{\mathcal{A}}$.
**Initialize:** initialize maximum turns $T \leftarrow \text{len}(\mathcal{C})//s$, devide the whole context $\mathcal{C}$ into $T$ chunks $\{\mathcal{C}_1, \cdots, \mathcal{C}_T\}$, $t \leftarrow 1, \mathcal{M}_0 \leftarrow \text{None}$.
**while** $t \leq \mathbf{T}$ **do**

   //get the update status, candidate memory, and exit status
   $\mathcal{U}_t, \hat{\mathcal{M}}_t, \mathcal{E}_t = \phi_\theta(\mathcal{Q}, \mathcal{C}_t, \mathcal{M}_{t-1})$
   **if** $\mathcal{U}_t$ == True **then**
     $\mathcal{M}_t \leftarrow \hat{\mathcal{M}}_t$       // update the memory
   **else**
     $\mathcal{M}_t \leftarrow \mathcal{M}_{t-1}$    // adopt the previous memory
   **end if**
   //(optional) exit based on the exit status
   **if** use_exit_gate == True and $\mathcal{E}_t$ == True **then**
     **break**
   **end if**               **GRU-Mem**
   $t \leftarrow t + 1$
**end while**
//answer the question based on the final memory
$\hat{\mathcal{A}} = \psi_\theta(\mathcal{Q}, \mathcal{M}_t)$
**return** $\hat{\mathcal{A}}$.

---

the loop. Once terminated, the final memory $\mathcal{M}_t$ will be sent into the answer agent $\psi_\theta$ immediately for answering the question. The detailed process of GRU-Mem is described in Algorithm 1.

### 3.2. Workflow Optimization with End-to-End RL

To teach the memory agent $\phi_\theta$ when to correctly activate the update gate and the exit gate, we explicitly reward correct gate status generation behaviors, beyond merely rewarding the final answer correctness. In this section, we first introduce the reward design in Section 3.2.1. After that, we discuss how to combine these rewards for the advantage calculation in Section 3.2.2. More training details can be found in the Appendix B.

#### 3.2.1. REWARD DESIGN

**Outcome reward.** For the outcome reward, we adopt the same reward as introduced in Equation 6. We assign the identical outcome reward $r_g^{\text{outcome}}$ for all the conversations $(o_{g,1}, o_{g,2}, ..., o_{g,T_g})$ within the whole trajectory in group $g$.

**Update reward.** To learn correctly activating the update gate, for each conversation at step $t$, we reward for generating the correct update gate status $\mathcal{U}_t$. Specifically, for chunks that contain evidence, the memory agent is rewarded when generating `<check>yes</check>` (*i.e.,* $\mathcal{U}_t$ == True),

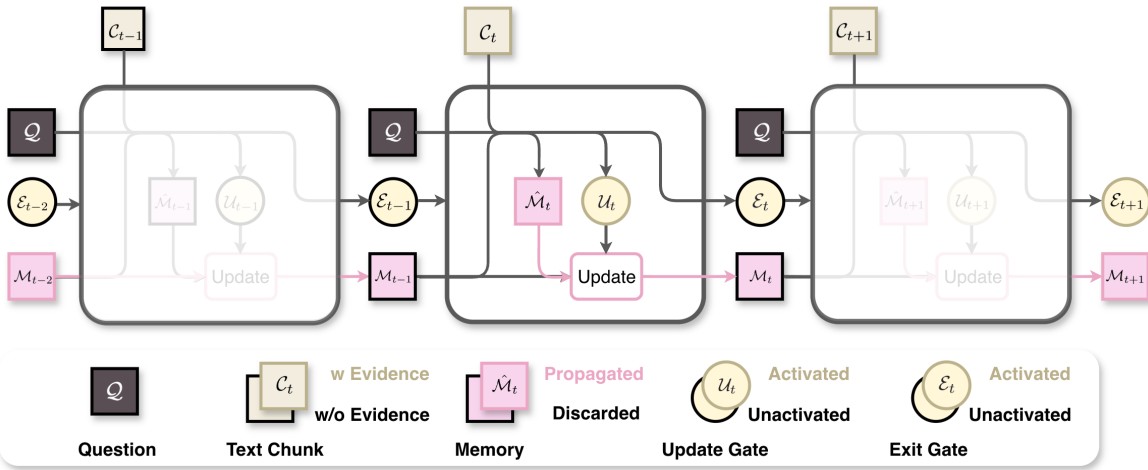

*Figure 3.* The memory updating process with the gated recurrent memory (GRU-Mem). At each time step $t$, the memory agent $\phi_\theta$ decides: (1) whether to update the memory $\mathcal{M}_t$ with the candidate memory $\hat{\mathcal{M}}_t$ or to keep the previous memory $\mathcal{M}_{t-1}$ unchanged, based on the update gate status $\mathcal{U}_t$ (True for updating and False for retaining $\mathcal{M}_{t-1}$); and (2) whether to stop scanning further chunks, based on the exit gate status $\mathcal{E}_t$ (True for exiting and False for continuing).

whereas for chunks without evidence, it is rewarded when generating `<check>no</check>` (*i.e.*, $\mathcal{U}_t ==$ False). This reward at step $t$ can be formulated as follows:

$$r_t^{\text{update}} = \begin{cases} 1, & \mathcal{U}_t \text{ is correct} \\ -1, & \mathcal{U}_t \text{ is incorrect.} \end{cases} \tag{8}$$

**Exit reward.** To learn the exit gate, we reward the whole trajectory when exiting at the correct position (*i.e.*, the turn that contains the last evidence required for answering the question, namely $t_{\text{last evidence}}$). This means the memory agent is required to generate `<next>end</next>` (*i.e.*, $\mathcal{U}_t ==$ True) when recognizing the occurrence of the last evidence, and generate `<next>continue</next>` (*i.e.*, $\mathcal{U}_t ==$ False). In other cases, all the conversations within the trajectory get punished. This reward is formulated as follows:

$$r^{\text{exit}} = \begin{cases} -0.75, & t_{\text{exit}} < t_{\text{last evidence}} \\ 0, & t_{\text{exit}} = t_{\text{last evidence}} \\ -0.5, & t_{\text{exit}} > t_{\text{last evidence}}. \end{cases} \tag{9}$$

Here, the $t_{\text{exit}}$ denotes the turn that the memory agent $\phi_\theta$ decides to exit the workflow. This reward design means the correct exit gate status $\mathcal{E}_t$ generation behavior does not get punished, and an early exit behavior gets more punishment than a late exit behavior due to the evidence insufficiency.

**Format reward.** To ensure that the generation $o_{g,t}$ of the memory agent $\phi_\theta$ can be parsed correctly, we introduce an additional format reward $r_{\text{format}}$. We checks whether the format meets the requirement of enclosed sequence of `<think> </think>`, `<check> </check>`, `<update> </update>`, and `<next> </next>`. Additionally, the content between `<check>` and `</check>` must be "yes" or "no", and the content between `<next>`

and `</next>` must be "continue" or "end". We make this reward strict so that only when all the generation output $(o_{g,1}, o_{g,2}, ..., o_{g,T_g})$ meets the format correctness, they will get a reward of 1. Otherwise, all the generation outputs get a reward of 0. This strict design is because we can not infer whether the incorrect format is caused by the previous erroneous parsing. This reward is formulated as:

$$r^{\text{format}} = \begin{cases} 1, & \text{The format of all turns is correct} \\ 0, & \text{Otherwise.} \end{cases} \tag{10}$$

**The overall trajectory-level reward.** Since all the generation outputs $(o_{g,1}, o_{g,2}, ..., o_{g,T_g})$ in the group $g$ share the same outcome reward, exit reward, and format reward, they can be combined as one total trajectory-level reward $r_g^{\text{traj}}$ as follows:

$$r_g^{\text{traj}} = r_g^{\text{outcome}} + r_g^{\text{exit}} + r_g^{\text{format}}. \tag{11}$$

### 3.2.2. ADVANTAGE CALCULATION

We calculate the trajectory-level advantage $\hat{A}_{g,t,i}^{\text{traj}}$ and the turn-level advantage $\hat{A}_{g,t,i}^{\text{turn}}$ respectively (Shi et al., 2025). After that, we combine them with a hyperparameter $\alpha$ for more flexible control. This is inspired by recent works of disentangling advantage calculation for stabilizing the training with different rewards (Shi et al., 2025; Liu et al., 2026). Here, the two advantages are calculated as follows:

$$\hat{A}_{g,t,i}^{\text{traj}} = r_g^{\text{traj}} - \frac{1}{G}\sum_{g=1}^{G} r_g^{\text{traj}}, \ \hat{A}_{g,t,i}^{\text{turn}} = r_{g,t}^{\text{update}} - \frac{1}{G_t}\sum_{g=1}^{G_t} r_{g,t}^{\text{turn}}. \tag{12}$$

As shown in Figure 4, the trajectory-level advantage $\hat{A}_{g,t,i}^{\text{traj}}$ is calculated between the **trajectories** of different groups. And the turn-level advantage $\hat{A}_{g,t,i}^{\text{turn}}$ is calculated between

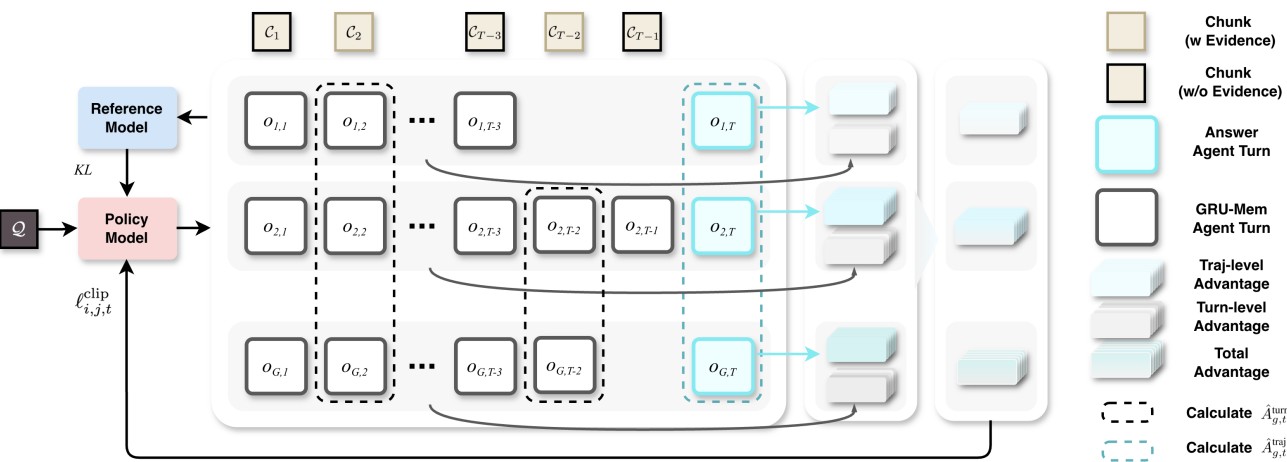

*Figure 4.* The advantage calculation process. The trajectory-level advantage $\hat{A}_{g,t}^{\text{traj}}$ and the turn-level advantage $\hat{A}_{g,t}^{\text{turn}}$ are calculated separately. They are combined into the total advantage with $\alpha$ (*i.e.*, $\hat{A}_{g,t,i} = \alpha\hat{A}_{g,t,i}^{\text{traj}} + (1-\alpha)\hat{A}_{g,t,i}^{\text{turn}}$).

turns at step $t$ from different groups. For the turn-level advantage $\hat{A}_{g,t,i}^{\text{turn}}$ calculation, the group size $G_t$ at step $t$ can be different from the trajectory group size $G$, since the workflow may exit at different step $T_g$ under the control of the exit gate. Therefore, these early exited groups have a smaller $T_g$ than others.

Finally, the total advantage $\hat{A}_{g,t,i}$ in Equation (3) is a combination of these two terms:

$$\hat{A}_{g,t,i} = \alpha\hat{A}_{g,t,i}^{\text{traj}} + (1-\alpha)\hat{A}_{g,t,i}^{\text{turn}}, \qquad (13)$$

where $\alpha$ is a hyperparameter for balancing two terms. The policy model will be optimized with the loss $\mathcal{J}(\theta)$, which is based on the calculated advantage $\hat{A}_{g,t,i}$.

### 3.3. Inference

Despite being always trained under the control of the exit gate. That is, during training, once $\mathcal{E}_t == \texttt{True}$, the workflow will exit immediately for answering the question. However, for some questions, we can not judge whether the evidence has been sufficient for answering the question since answering the question requires reading the whole context (*e.g.,* "What are *all* the special magic numbers for xxx" as introduced in the multi-values task in the RULER (Hsieh et al., 2024) benchmark). Therefore, we provide two kinds of inference strategies: with the exit gate (w EG) and without the exit gate (w/o EG), for the inference flexibility. For the w/o EG inference mode, the generation workflow would not exit even when $\mathcal{E}_t == \texttt{True}$.

## 4. Experiments

In this section, we explore the effectiveness of GRU-Mem by answering the following questions.

- **RQ1:** (Performance and Efficiency) Can GRU-Mem achieve better performance compared to the vanilla MemAgent with a higher inference efficiency?

- **RQ2:** (Gating Mechanism) How do the introduced two gates impact the workflow behaviors?

- **RQ3:** (Ablation Study) How does the hyperparameter $\alpha$ affect the training dynamics of GRU-Mem? How does the RL training contribute to the final performance?

**LLMs.** We conduct experiments on two open-source LLM backbones with different sizes: Qwen2.5-3B-Instruct and Qwen2.5-7B-Instruct (Yang et al., 2024). We train these LLMs on the same data as introduced in MemAgent.

**Evaluation Benchmarks.** We strictly follow the evaluation setup as introduced in MemAgent (Yu et al., 2025a). Specifically, we evaluate methods on one multi-hop QA task hoppotqa (HQA) (Yang et al., 2018), one single-hop QA task SQuAD (Rajpurkar et al., 2016), three single-key NIAH tasks (*i.e.,* SK-1, SK-2, SK-3), three multi-key NIAH tasks (*i.e.,* MK-1, MK-2, MK-3), one multi-queries NIAH task MQ, and one multi-values task MV. Here, the number -N indicates the difficulty level, with larger values corresponding to greater difficulty. HQA is one in-distribution task similar to the training data, while the remaining tasks are out-of-distribution tasks. The benchmark preparation strictly follows those in the MemAgent (Yu et al., 2025a). These tasks are constructed with different context lengths, ranging from 7K to 896K.

More details about the training and evaluation can be found in the Appendix B.

*Table 1.* The performance comparison across diverse long-context tasks.

| Scale | Method | Avg. Metric | | Tasks | | | | | | | | |
|---|---|---|---|---|---|---|---|---|---|---|---|---|
| | | | HQA | SQuAD | SK-1 | SK-2 | SK-3 | MK-1 | MK-2 | MK-3 | MQ | MV |
| 7B | MemAgent | Perf. % ↑ | 76.07 | 79.56 | 99.78 | 95.54 | **97.66** | 97.21 | 75.78 | **95.98** | 88.37 | 81.70 |
| | | Time s ↓ | 463.38 | 162.34 | 378.51 | 420.38 | 419.74 | 412.93 | 349.61 | 403.60 | 419.53 | 407.89 |
| | GRU-Mem (w/o EG) | Perf. % ↑ | 75.59 | **80.73** | **100.00** | 95.43 | 95.98 | 98.10 | 67.52 | 93.53 | **96.43** | **95.23** |
| | | Time s ↓ | 284.41 | 85.03 | 135.60 | 171.57 | 154.03 | 168.54 | 258.40 | 242.35 | 156.62 | **153.08** |
| | GRU-Mem (w EG) | Perf. % ↑ | **76.37** | 80.47 | **100.00** | **96.65** | 95.20 | **98.55** | **84.15** | 95.54 | 84.12 | - |
| | | Time s ↓ | **209.33** | **64.32** | **126.00** | **113.89** | **107.64** | **102.46** | **123.53** | **136.02** | **108.62** | - |
| 3B | MemAgent | Perf. % ↑ | 63.87 | 67.58 | **96.76** | 88.73 | 86.72 | 79.46 | 35.05 | 44.42 | **77.26** | 36.27 |
| | | Time s ↓ | 218.60 | 68.81 | 122.31 | 176.48 | 182.72 | 146.77 | 118.16 | 165.48 | 177.22 | 154.78 |
| | GRU-Mem (w/o EG) | Perf. % ↑ | **69.04** | **69.92** | 94.42 | **88.84** | 89.40 | **91.52** | 67.08 | **91.41** | 73.91 | **59.46** |
| | | Time s ↓ | 211.77 | 60.71 | 114.49 | 120.89 | 118.02 | 116.92 | 140.82 | 122.88 | 121.32 | 118.71 |
| | GRU-Mem (w EG) | Perf. % ↑ | 65.33 | 69.66 | 95.31 | 88.28 | **90.85** | 89.84 | 58.15 | 90.85 | 63.84 | – |
| | | Time s ↓ | **162.31** | **45.30** | **104.38** | **84.29** | **82.13** | **74.19** | **59.60** | **68.46** | **80.29** | – |

## 4.1. (RQ1) Performance and Efficiency Comparison

We report the averaged performance and inference time cost across different context sizes in Table 1, and the averaged performance and time across different context sizes of a single task in Figure 5. More results under varying context lengths can be found in Appendix D.1. We have the following observations:

- **GRU-Mem generally outperforms the vanilla MemAgent across diverse datasets.** As shown in Table 1, under both w/o EG and w EG inference modes, GRU-Mem outperforms MemAgent on most of the tasks. Moreover, compared to MemAgent, GRU-Mem excels at out-of-distribution tasks, as it performs much better on NIAH tasks. Additionally, under the backbone LLM size of 3B, GRU-Mem even gains more. For example, GRU-Mem consistently maintains a high performance on MK task series, while MemAgent shows a sharp performance drop. We attribute the success of GRU-Mem to its more stable memory updating with the introduced update gate.

- **GRU-Mem exhibits superior inference efficiency compared to the vanilla MemAgent.** As shown in the row of Time in Table 1, under both w/o EG and w EG inference modes, the GRU-Mem exhibits significant inference efficiency improvements compared to MemAgent. Under the w/o EG mode, GRU-Mem generally achieves around 200% inference acceleration. Under the w EG mode, when the early exit mechanism is activated, the inference acceleration can be even faster, achieving a maximum 400% times faster in several cases, such as MK-1, without harming the performance.

## 4.2. (RQ2) Study of Gating Mechanisms

In this section, we study how the two introduced gating mechanisms affect the behaviors of the recurrent workflow.

**Update Gate.** We study how the update gate reduces the risk of memory explosion by tracking the memory size dynamics

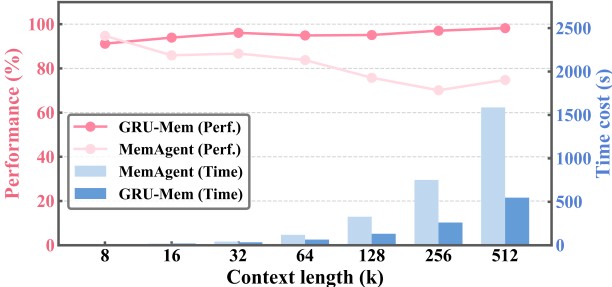

*Figure 5.* Performance and efficiency across diverse context lengths on the MV task.

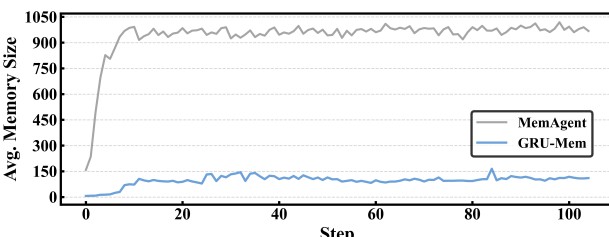

*Figure 6.* Memory size dynamics on MV task (512K context size).

during the long-term inference. As shown in Figure 6, GRU-Mem shows a much lower memory size increasing speed, while MemAgent quickly encounters the memory explosion when the size of memory reaches the maximum memory size of 1024 tokens. This phenomenon is because GRU-Mem only updates the memory on a few critical chunks, while MemAgent may indiscriminately update the memory. As a result, in MemAgent, the ever-growing memory hurts its performance, and generating overly long memory also significantly increases inference overhead.

**Exit Gate.** For the exit gate, we especially focus on how it can benefit the inference when the last evidence may occur very early, with possible reranking techniques. To simulate this, we manually construct an unbalanced evidence occurrence setting, where the last evidence must occur at the top 20% documents. We report the performance and efficiency comparison of the 7B-sized model under this setting in Table 2. As shown in this table, GRU-Mem (w EG) reduces the inference time to 1/4 of the vanilla MemAgent. We also additionally calculate the ratio of early stopping,

*Table 2.* Performance when evidence occurs at top 20% positions.

| Method | Metric | Context Length | | | |
|---|---|---|---|---|---|
| | | 112K | 224K | 448K | 896K |
| MemAgent | Perf. % ↑ | 79.69 | 78.91 | 78.12 | 80.47 |
| | Time s ↓ | 171.65 | 358.60 | 804.23 | 1691.93 |
| GRU-Mem (w EG) | Perf. % ↑ | 78.91 | 82.03 | 80.47 | 78.12 |
| | Time s ↓ | 60.81 | 111.67 | 213.04 | 454.72 |

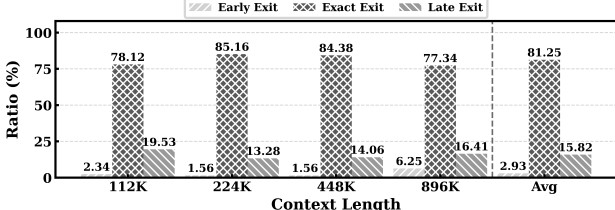

*Figure 7.* The ratio of early, exact, and late exit.

exact stopping, and late stopping in Figure 7. As shown, GRU-Mem can identify the last evidence position and exit accordingly in most cases. Such property makes GRU-Mem work more flexibly under evidence-unbalanced scenarios. More results are in Appendix D.3.

### 4.3. (RQ3) Ablation Study

In this section, we conduct ablation studies on two aspects: the impact of $\alpha$ selection on training dynamics, and RL training's improvement to workflow performance.

**Impact of $\alpha$.** We visualize the training dynamics under different selection of $\alpha$ (*i.e.,* 1.0, 0.9, and 0.5) in Figures 8 and 9. Figure 8 reflects the update accuracy on evidence-present (8a) and evidence-free (8b) chunks. Figure 9 shows the ratio of exactly stopping at the last evidence turn (9a) and the reward on the validation set (9b). We have the following observations: **A higher $\alpha$ improves accuracy on evidence-present chunks, but also increases the risk of unnecessary updating on evidence-free chunks.** As shown in Figure 8, when $\alpha = 1$ (*i.e.,* no reward for the update gate), the accuracy on evidence-free chunks drops dramatically, indicating that without the update reward, LLMs tend to update the memory indiscriminately. **A mild $\alpha$ leads to a balanced update accuracy on both evidence-present and evidence-free chunks.** Additionally, with an $\alpha$ of 0.9, the performance on the validation set shows a more stable trend, outperforming experiments with $\alpha$ values of 1.0 and 0.5. Moreover, across all $\alpha$ settings, the LLM effectively learns the exit behavior, achieving an exit accuracy over 0.8, as shown in Figure 9a. Based on the above, we adopt the $\alpha$ of 0.9 as the default setting due to the relatively high reward and balanced update gate accuracy. More results about the training dynamics can be found in Appendix D.2.

**Effectiveness of RL training.** We further report the effectiveness of RL training in Figure 10. As shown in this figure, RL training generally brings performance gain, especially on hard tasks such as HQA, SQuAD, and MK series.

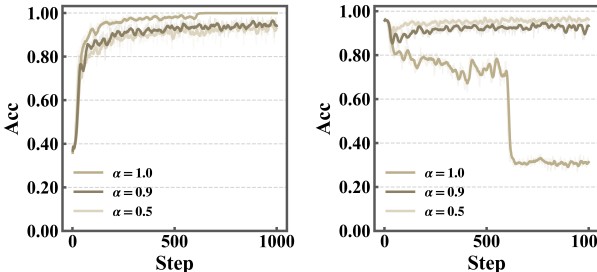

*(a)* On evidence-present chunks. *(b)* On evidence-free chunks.
*Figure 8.* The accuracy dynamics of the update gate status $\mathcal{U}_t$

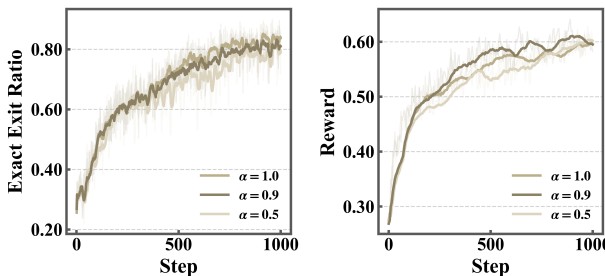

*(a)* The ratio of exactly exiting. *(b)* Reward on the validation set.
*Figure 9.* Training dynamics.

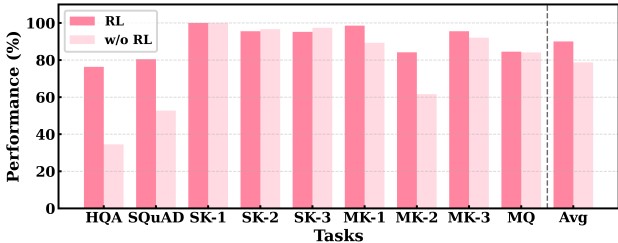

*Figure 10.* Effectiveness of RL training.

## 5. Limitations

Several limitations remain in this paper. On one hand, it is limited to the QA domain, with other tasks (e.g., summarization) largely underexplored. On the other hand, the extra rewards in GRU-Mem reduce training stability, requiring a smaller off-policy degree and longer convergence time.

## 6. Conclusion

While recent recurrent memory work MemAgent (Yu et al., 2025a) provided a chunk-by-chunk paradigm for addressing long-context reasoning, it suffers from memory explosion due to indiscriminate updates and wasted computation from lacking an early exit mechanism. To address these issues, we proposed GRU-Mem, which equips the recurrent loop with two text-controlled gates: an update gate that updates memory only when necessary and an exit gate that terminates once sufficient evidence is collected, which are trained end-to-end with two reward signals $r^{\text{update}}$ and $r^{\text{exit}}$. Experiments across diverse long-context reasoning tasks showed GRU-Mem outperforms vanilla MemAgent while achieving up to 400% inference speed acceleration.

## Impact Statement

This paper presents a study aimed at advancing long-context reasoning via a memory mechanism, a work that offers notable benefits to domains such as long-document question answering. This work mainly brings significant positive societal effects, for instance, helping humans read and process long contexts more efficiently and quickly. The negative impacts are largely trivial in comparison, and one potential issue is that LLM hallucinations in long-context reading may lead humans to make wrong decisions.

## Acknowledgement

This research is supported by National Natural Science Foundation of China (U25A20445) and the National Research Foundation, Singapore under its National Large Language Models Funding Initiative, (AISG Award No: AISG-NMLP-2024-002). Any opinions, findings and conclusions or recommendations expressed in this material are those of the author(s) and do not reflect the views of National Research Foundation, Singapore.

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

# A. Related Works

## A.1. Long-context Reasoning

Reasoning over long contexts remains challenging for large language models (LLMs) (Bai et al., 2024; Hsieh et al., 2024), while this capability is important for various downstream tasks such as reading a whole book (Yu et al., 2025a). Existing works show that LLM performance often degrades when relevant evidence is dispersed across long sequences, particularly when key information appears in the middle of the input, which is commonly referred to as lost in the middle (Liu et al., 2024; 2025b). Traditional methods for long-context reasoning can be broadly categorized into architectural modifications (Child et al., 2019; Katharopoulos et al., 2020; Gu & Dao, 2023) and context extension (Su et al., 2024b; Peng et al., 2024) techniques. For architectures, sparse attention mechanisms restrict attention patterns using predefined structures such as sliding windows or global tokens to reduce computational cost (Beltagy et al., 2020; Zaheer et al., 2020), while linear attention methods approximate softmax attention to achieve linear-time complexity (Katharopoulos et al., 2020). For context extension methods, they typically focus on positional embedding extrapolation, such as RoPE (Su et al., 2024b) and YaRN (Peng et al., 2024). While showing effectiveness in addressing long-context reasoning, such traditional methods still suffer from performance degradation in overly long contexts.

## A.2. LLM Memory

To address the limitations that LLMs can only process limited information within their context window size, recent works have begun to explore augmenting the LLM with memory mechanisms (Packer et al., 2023; Li et al., 2025). Such memory mechanisms have also been utilized for long-context reasoning recently (Yu et al., 2025a; Li et al., 2025). The key idea is to read the long context chunk-by-chunk, and recurrently maintain a textual memory, and answer the question with the final memory. With end-to-end reinforcement learning, such techniques enable the LLM to significantly outperform the feed-in-once long-context reasoning paradigm (Xiong et al., 2024; Bai et al., 2024). While effective, such methods face risks of memory explosion and a lack of exiting mechanism.

## A.3. Reinforcement Learning with Multiple Tasks

Recent efforts have demonstrated that LLMs can learn multiple objectives (Zhao et al., 2025; Liu et al., 2025a; Lu et al., 2025) by rewarding different learning targets. On the one hand, a single LLM can learn to play multiple roles that compete with one another, such as acting as both a reasoner and a data generator (Zhao et al., 2025; He et al., 2025), where the reward of one role is the loss of another. On the other hand, a single LLM can also learn to collaborate across roles, for example, by completing a safety alignment task (Zhang et al., 2025a) or performing long-context reasoning (Yu et al., 2025a; Shi et al., 2025), where the reward comes from the total task completion. These advances inspire us to design role-specific rewards for a single policy model, enabling it to learn both update and exit behaviors for long-context reasoning.

## B. Implementation Details

We conduct all the experiments based on the verl [1] framework. For the RL training, we train all models with a chunk size of 5,000 tokens, a maximum prompt length of 8,192 tokens, and a maximum response length of 2,048 tokens. We adopt a clip ratio of 0.2 and set the learning rate to $1 \times 10^{-6}$. During training, we sample responses with temperature $1.0$ and top-$p$ $1.0$, while for validation we use temperature $1.0$ and top-$p$ $0.7$. The training batch size is set to 128, with rollout number $N = 16$ and PPO mini-batch size 128. We apply a learning rate warmup for 20 steps. We stop training until we observe the convergence of reward on the validation set. All the evaluations are conducted on an 8-GPU node.

| Hyperparameter | Value |
|---|---|
| Chunk size | 5000 |
| Max prompt length | 8192 |
| Max response length | 2048 |
| Clip ratio | 0.20 |
| Learning rate | $1 \times 10^{-6}$ |
| Sampling temperature (train) | 1.0 |
| Top_p (train) | 1.0 |
| Sampling temperature (val) | 1.0 |
| Top_p (val) | 0.7 |
| Train batch size | 128 |
| Rollout number ($N$, train) | 16 |
| Mini batch size | 128 |
| LR warmup steps | 20 |

*Table 3.* Key training hyperparameters used in RL training.

---

[1]https://github.com/volcengine/verl

## C. Prompts

We present the prompts adopted for the memory agent $\phi_\theta$ and answer agent $\psi_\theta$ in GRU-Mem in Figure 11a and Figure 11b, respectively.

---

**Prompt:**

```
You are presented with a problem, a section of an article that may contain the
answer to the problem, and a previous memory.  Please read the provided section
carefully.  You should reason about whether the new section contains useful
information about the problem, and then update the memory with the new
information that helps to answer the problem.
Be sure to retain all relevant details from the previous memory while adding
any new, useful information.  You should also carefully judge whether you have
collected enough information to answer the problem.
You should reason about whether the new section contains useful information,
what to update, and what to do next first between <think> and </think>.
If the new section contains useful information about the problem, you should
first generate <check>yes</check>.  After that, update the new memory between
<update> and </update>.
If the new section does not contain useful information about the problem, you
should first generate <check>no</check>.  After that, you should keep the
previous memory unchanged between <update> and </update>.
In the end, if you haven't collected enough information for the problem, return
<next>continue</next>.  ONLY when enough information is collected, return
<next>end</next>.
<problem> {prompt} </problem>
<memory> {memory} </memory>
<section> {chunk} </section>
```

*(a)* Prompt for memory agent $\phi_\theta$ in GRU-Mem.

---

**Prompt:**

```
You are presented with a problem and a previous memory.  Please answer the
problem based on the previous memory and put the answer in boxed .
<problem> {prompt} </problem>
<memory> {memory} </memory>
Your answer:
```

*(b)* Prompt for answer agent $\psi_\theta$ in GRU-Mem.

*Figure 11.* Prompts for memory and answer agents.

# D. Experiments

## D.1. Performance Under Varying Context Lengths

We present the model's performance and efficiency across a range of context lengths in Figures 12 to 21. In these figures, numerical values represent performance metrics, while color shading indicates the acceleration ratio, where deeper hues corresponding to a higher acceleration ratio. As shown, GRU-Mem consistently yields a higher inference speed, with the acceleration more obvious as the context length increases.

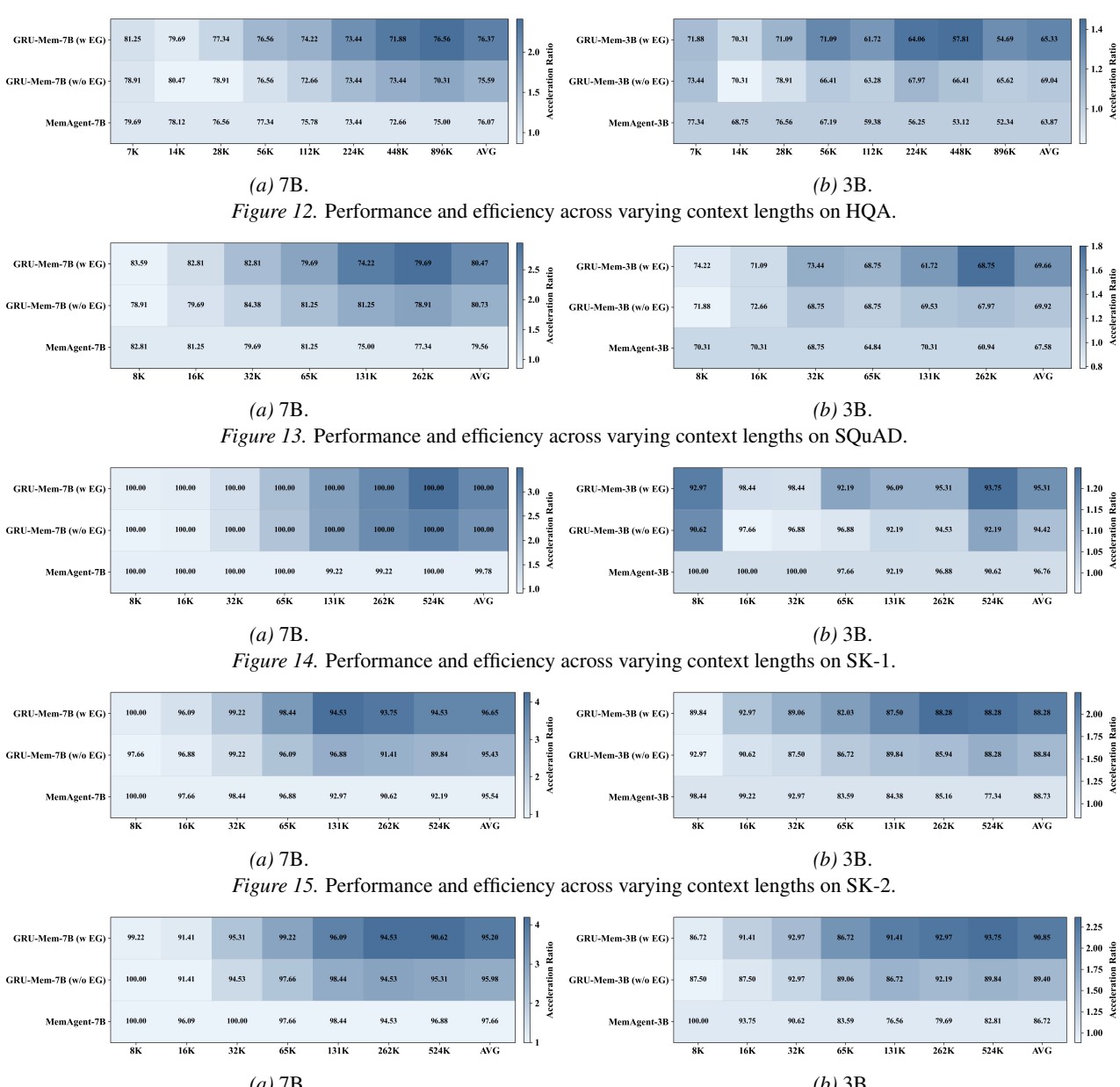

*(a)* 7B.  *(b)* 3B.

*Figure 12.* Performance and efficiency across varying context lengths on HQA.

*(a)* 7B.  *(b)* 3B.

*Figure 13.* Performance and efficiency across varying context lengths on SQuAD.

*(a)* 7B.  *(b)* 3B.

*Figure 14.* Performance and efficiency across varying context lengths on SK-1.

*(a)* 7B.  *(b)* 3B.

*Figure 15.* Performance and efficiency across varying context lengths on SK-2.

*(a)* 7B.  *(b)* 3B.

*Figure 16.* Performance and efficiency across varying context lengths on SK-3.

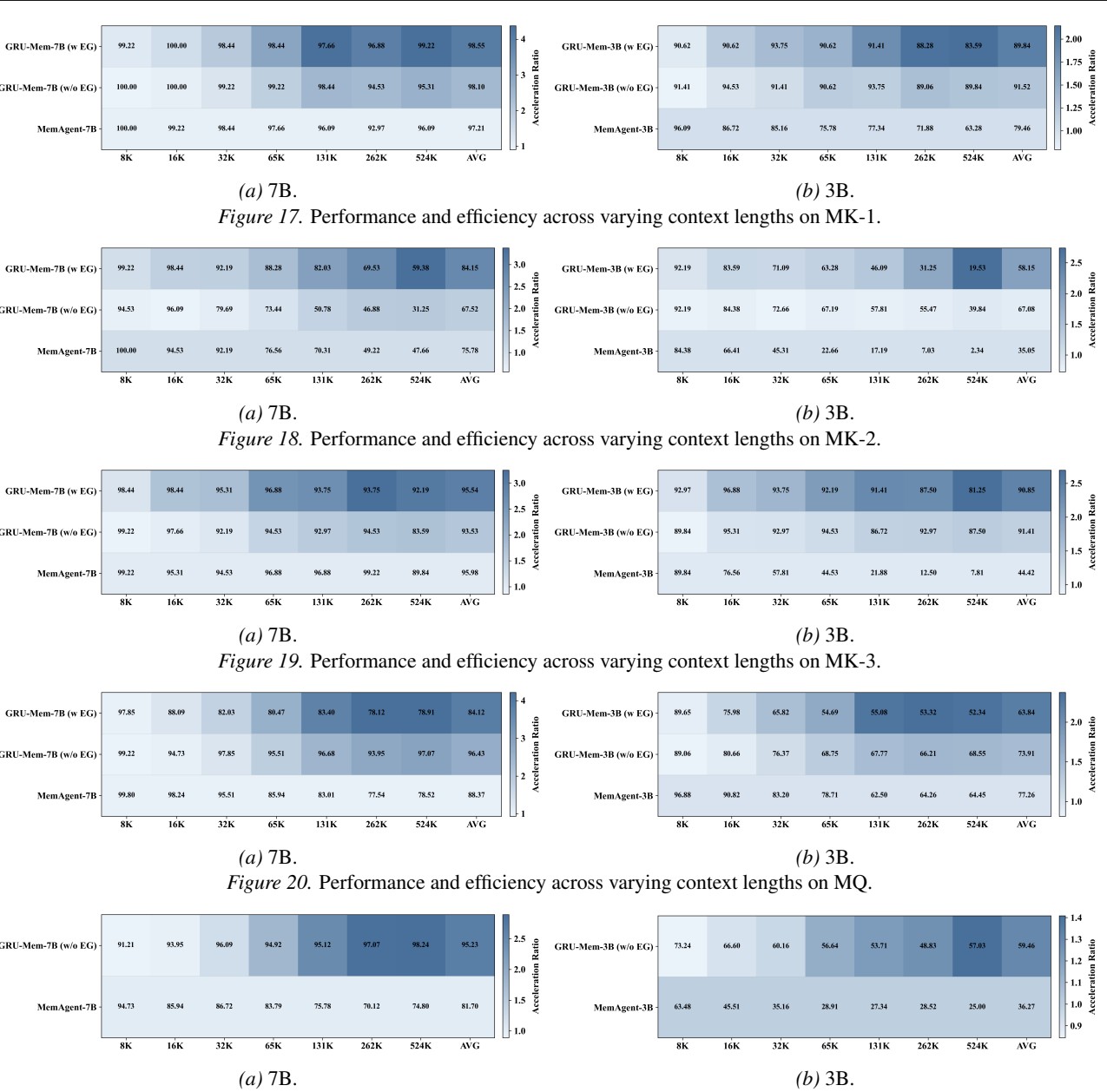

*(a)* 7B.      *(b)* 3B.

*Figure 17.* Performance and efficiency across varying context lengths on MK-1.

*(a)* 7B.      *(b)* 3B.

*Figure 18.* Performance and efficiency across varying context lengths on MK-2.

*(a)* 7B.      *(b)* 3B.

*Figure 19.* Performance and efficiency across varying context lengths on MK-3.

*(a)* 7B.      *(b)* 3B.

*Figure 20.* Performance and efficiency across varying context lengths on MQ.

*(a)* 7B.      *(b)* 3B.

*Figure 21.* Performance and efficiency across varying context lengths on MV.

## D.2. Training Dynamics

We report more details about the training dynamics under different $\alpha$ in this section, including the format correctness (Figure 22a), the average response length (Figure 22b), the absolute exit deviation (Figure 23a), the exit deviation (Figure 23b), the ratio of early exit (Figure 24b), and the ratio of late exit (Figure 24c). We have the following observations:

- **Correct formatting is rapidly acquired.** As shown in Figure 22a, under all the settings of $\alpha$, the format correctness quickly reach around 100% accuracy with only a few steps.

- **The introduction of update gate reduces the increase of response length.** As shown in Figure 22b, a higher $\alpha$ exhibits a sharper response length increasing trend. This is because the introduction of the update gate encourage the memory agent only to update on evidence-present chunks, which reduces unnecessary updates on evidence-free chunks and thus lowers the average response length.

- **The correct exit behavior is gradually learned.** As shown in Figure 23a and Figure 23b, the average absolute exit deviation (*i.e.,* $|t_{\text{exit}} - t_{\text{last evidence}}|$) decreases stably across different settings of $\alpha$, and the exit deviation (*i.e.,* $t_{\text{exit}} - t_{\text{last evidence}}$) also converges to near zero. Additionally, the ratio of exactly exiting at the correct place also increases stably (*i.e.,* Figure 24a), while the ratio of early exit and late exit decreases accordingly (*i.e.,* Figure 24b and Figure 24c).

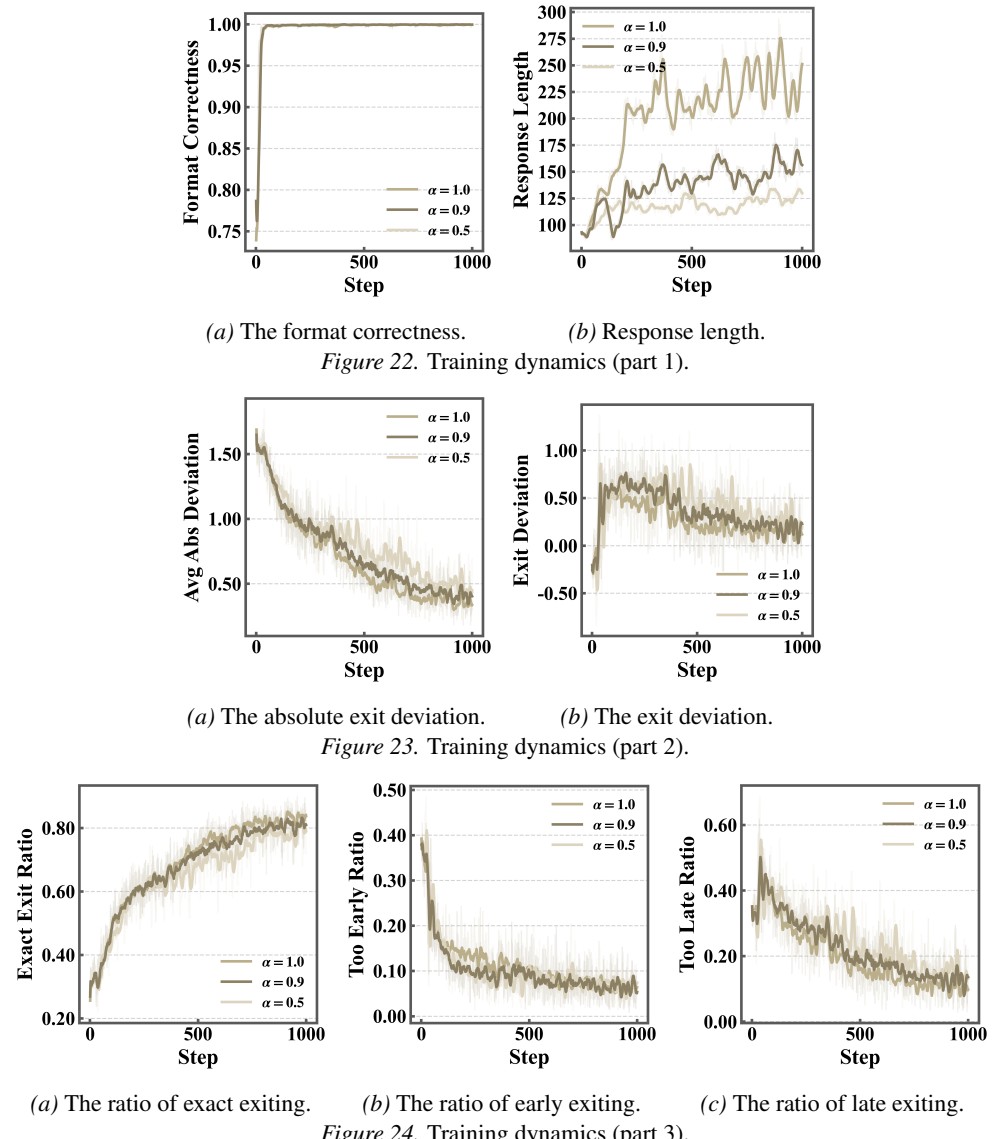

*(a)* The format correctness.  *(b)* Response length.

*Figure 22.* Training dynamics (part 1).

*(a)* The absolute exit deviation.  *(b)* The exit deviation.

*Figure 23.* Training dynamics (part 2).

*(a)* The ratio of exact exiting.  *(b)* The ratio of early exiting.  *(c)* The ratio of late exiting.

*Figure 24.* Training dynamics (part 3).

### D.3. Performance Under Unbalanced Evidence Distribution

We further test the performance and exit accuracy when evidence only occurs at the top 10% position. We report the performance and efficiency in Table 4. We also report the ratio of early exit, exact exit, and late exit in Figure 25. As shown, under the unbalanced evidence occurrence distribution, where the last evidence must occur at the top 10% documents, GRU-Mem also shows an accurate ratio of exiting at the correct place (*i.e.,* around 80%). With this accurate exiting ratio, GRU-Mem exhibits much faster inference time acceleration, while maintaining the same performance.

*Table 4.* Performance when evidence occurs at top 10% positions.

| Method | Metric | Context Length | | | |
| --- | --- | --- | --- | --- | --- |
| | | 112K | 224K | 448K | 896K |
| MemAgent | Perf. % ↑ | 78.12 | 76.56 | 79.69 | 78.91 |
| | Time s ↓ | 178.13 | 373.26 | 799.81 | 1652.31 |
| GRU-Mem (w EG) | Perf. % ↑ | 78.12 | 81.25 | 80.47 | 78.12 |
| | Time s ↓ | 62.65 | 102.48 | 205.15 | 405.54 |

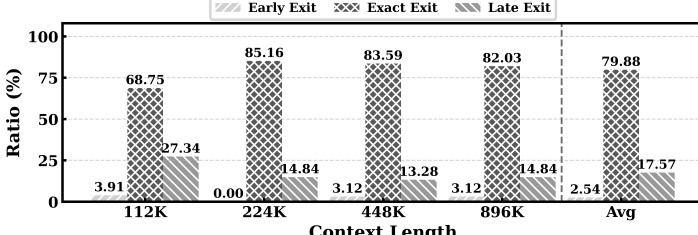

*Figure 25.* The ratio of early, exact, and late exit.

# E. Case Study

In this section, we present several cases about how the memory agent in GRU-Mem works, together with the memory updating dynamics.

**Case 1:** What science fantasy young adult series, told in first person, has a set of companion books narrating the stories of enslaved worlds and alien species?

**Ground Truth:** Animorphs.

**Prediction:** Animorphs.

```
<think>The provided section does not contain any information about a science fantasy
young adult series or companion books narrating the stories of enslaved worlds and
alien species.  Therefore, there is no new useful information to add to the existing
memory.</think>
<check>no</check>
<update>No previous memory</update>
<next>continue</next>
```

*Figure 26.* Response at turn 65 (evidence-free chunk).

```
<think>The new section mentions Änimorphs,̈which is described as a science fantasy
series for young adults.  It is told in first person and involves multiple
narrators.  The text also describes the core themes of the series, but does not
mention anything about companion books or enslaved worlds.  We need to check if this
series has companion books that narrate the stories of enslaved worlds and alien
species.</think>
<check>yes</check>
<update>Animorphs is a science fantasy series of young adult books written by
Katherine Applegate and her husband Michael Grant, writing together under the name
K. A. Applegate, and published by Scholastic.  It is told in first person, with all
six main characters taking turns narrating the books through their own perspectives.
Horror, war, dehumanization, sanity, morality, innocence, leadership, freedom and
growing up are the core themes of the series.</update>
<next>continue</next>
```

*Figure 27.* Response at turn 66 (evidence-present chunk).

```
<think>The new section provides information about T̈he Hork-Bajir Chronicles,̈ which
is a companion book to the Änimorphs̈series.  It mentions that T̈he Hork-Bajir
Chronicles̈takes place before T̈he Pretenderänd narrates the story of the Hork-Bajir
and their enslavement.  This seems relevant to the question, as it suggests that the
Änimorphs̈series has companion books that tell the story of an enslaved species,
which could potentially answer the problem.</think>
<check>yes</check>
<update>Animorphs is a science fantasy series of young adult books written by
Katherine Applegate and her husband Michael Grant, writing together under the name
K. A. Applegate, and published by Scholastic.  It is told in first person, with all
six main characters taking turns narrating the books through their own perspectives.
T̈he Hork-Bajir Chronicles̈is the second companion book to the **Änimorphs**̈ series,
written by K. A. Applegate.  With respect to continuity within the series, it takes
place before book #23, T̈he Pretender,̈ although the events told in the story occur
between the time of T̈he Ellimist Chroniclesänd T̈he Andalite Chronicles.̈  The book is
introduced by Tobias, who flies to the valley of the free Hork-Bajir, where Jara
Hamee tells him the story of how the Yeerks enslaved the Hork-Bajir, and how Aldrea,
an Andalite, and her companion, Dak Hamee, a Hork-Bajir, tried to save their world
from the invasion.  Jara Hamee's story is narrated from the points of view of
Aldrea, Dak Hamee, and Esplin 9466, alternating in similar fashion to the
M̈egamorphs̈books.</update>
<next>**end**</next>
```

*Figure 28.* Response at turn 74 (evidence-present & last evidence chunk).

**Case 2:** What is the inhabitant of the city where 122nd SS-Standarte was formed in 2014?

**Ground Truth:** 276,170 inhabitants.

**Prediction:** 276,170.

```
<think>The provided section does not mention anything related to the 122nd
SS-Standarte or the city where it was formed in 2014.  Therefore, there is no useful
information here to address the problem.</think>
<check>no</check>
<update>No previous memory</update>
<next>continue</next>
```

*Figure 29.* Response at turn 15 (evidence-free chunk).

```
<think>The new section contains information about the 122nd SS-Standarte and its
formation in Strasbourg.  This directly relates to the problem which asks about the
inhabitant of the city where the 122nd SS-Standarte was formed in 2014.  The new
information is relevant.</think>
<check>yes</check>
<update>The 122nd SS-Standarte was a regimental command of the Allgemeine-SS that
was formed in the city of Strasbourg during World War II. The Standarte was
activated on November 12, 1940, and reached battalion strength by the end of the
year.  The command was a successor to the previously disbanded 121st SS-Standarte,
also situated in Strasbourg.</update>
<next>continue</next>
```

*Figure 30.* Response at turn 16 (evidence-present chunk).

```
<think>The new section provides detailed information about Strasbourg, stating that
in 2014, the city proper had 276,170 inhabitants and that both the Eurométropole de
Strasbourg and the Arrondissement of Strasbourg had 484,157 inhabitants.  The
section also mentions that Strasbourg is the capital and largest city of the Grand
Est region of France.  Given that the original passage stated that the 122nd
SS-Standarte was formed in Strasbourg during World War II, this new information
directly addresses the problem by identifying the city and its current
population.</think>
<check>yes</check>
<update>Strasbourg ( , ] ; Alsatian:  Štrossburiˑ German:  ŠtraßburgÏ ) is the
capital and largest city of the Grand Est region of France and is the official seat
of the European Parliament.  Located close to the border with Germany in the
historic region of Alsace, it is the capital of the Bas-Rhin département.  In 2014,
the city proper had **276,170 inhabitants** and both the Eurométropole de Strasbourg
(Greater Strasbourg) and the Arrondissement of Strasbourg had 484,157 inhabitants.
Strasbourg's metropolitan area had a population of 773,347 in 2013 (not counting the
section across the border in Germany), making it the ninth largest metro area in
France and home to 13% of the Grand Est region's inhabitants.  The transnational
Eurodistrict Strasbourg-Ortenau had a population of 915,000 inhabitants in 2014.
The 122nd SS-Standarte was a regimental command of the Allgemeine-SS that was formed
in the city of Strasbourg during World War II. The Standarte was activated on
November 12, 1940, and reached battalion strength by the end of the year.  The
command was a successor to the previously disbanded 121st SS-Standarte, also
situated in Strasbourg.</update>
<next>**end**</next>
```

*Figure 31.* Response at turn 16 (evidence-present chunk).

**Case 3:** A special magic uuid is hidden within the following text. Make sure to memorize it. What is the special magic uuid for belligerent-councilperson mentioned in the provided text?

**Ground Truth:** ac5f891e-ac5d-4d3b-b830-a7efe7cf807a.

**Prediction:** ac5f891e-ac5d-4d3b-b830-a7efe7cf807a.

```
<think>The new section does not mention any belligerent-councilpersonör a special
magic UUID. It discusses topics related to addiction, technology, and Apple's App
Store approval process.  Therefore, the new section does not provide useful
information to identify the special magic UUID for the
belligerent-councilperson.</think>
<check>no</check>
<update>No previous memory</update>
<next>continue</next>
```

*Figure 32.* Response at turn 1 (evidence-free chunk).

```
<think>The provided section does not mention any special magic UUID associated with
a belligerent-councilperson.  The text is about various topics including essay
writing, venture capital, and some historical context related to a startup.  No
specific UUID is referenced.</think>
<check>no</check>
<update>No previous memory</update>
<next>continue</next>
```

*Figure 33.* Response at turn 22 (evidence-free chunk).

```
<think>The new section does not contain any information related to a special magic
uuidör the context of belligerent-councilperson.  It discusses various aspects of
essay writing, startup culture, and Paul Graham's experiences.  Therefore, it does
not provide useful information to solve the problem.</think>
<check>no</check>
<update>No previous memory</update>
<next>continue</next>
```

*Figure 34.* Response at turn 28 (evidence-free chunk).

```
<think>The provided section does not mention a belligerent-councilpersonör any
special magic UUID. The content discusses startups, the internet economy, and the
changing dynamics between employers, investors, and young talent.  No information
related to the specific problem is present.</think>
<check>no</check>
<update>No previous memory</update>
<next>continue</next>
```

*Figure 35.* Response at turn 66 (evidence-free chunk).

```
<think>The new section contains information about a special magic UUID for a
belligerent-councilperson, which is relevant to the problem.  It provides the
specific UUID: ac5f891e-ac5d-4d3b-b830-a7efe7cf807a.</think>
<check>yes</check>
<update>One of the special magic uuids for belligerent-councilperson is:
ac5f891e-ac5d-4d3b-b830-a7efe7cf807a.  </update>
<next>end</next>
```

*Figure 36.* Response at turn 100 (evidence-present & last evidence chunk).

