# OpenReview forum: "When to Memorize and When to Stop: Gated Recurrent Memory for Long-Context Reasoning"
_ICML.cc/2026/Conference — ICML 2026 regular_

### Official Review · Reviewer_mBvY · 2026-02-18

**Soundness:** 3
**Presentation:** 2
**Significance:** 2
**Originality:** 2
**Overall Recommendation:** 4
**Confidence:** 4

**Summary:**

This paper explores how to help LLM extremely long contexts more efficiently and reliably. The authors conduct comprehensive evaluations across a range of datasets and context lengths. The reported 4× inference speedup is especially striking, making it highly valuable for real-world applications that involve long-context processing.

**Compliance With Llm Reviewing Policy:**

Affirmed.

**Final Justification:**

To align with the authors' statement, I  improve my score.

**Key Questions For Authors:**

This paper extends the MemAgent framework by introducing two gating mechanisms. While the empirical improvements are clear, we would like to better understand the motivation behind this architectural redesign. Specifically, what fundamental limitations in MemAgent made it necessary to introduce gates, rather than addressing the issues through prompt tuning or refining the reward function? In other words, why are gates essential? What core weaknesses of the RNN-style memory paradigm do they address that cannot be resolved through better training alone?

**Limitations:**

yes

**Strengths And Weaknesses:**

The idea of adapting GRU-style gating to address issues in RNN-like memory architectures is a natural extension. Its practical value lies in successfully adapting this idea to LLM-based long-context scenarios, where it delivers substantial gains in efficiency. That said, the work builds upon the foundation of prior work—specifically MemAgent. GRU-Mem can be seen as adding two gating mechanisms to MemAgent. While the improvements are notable, the core recurrent memory paradigm is not original to this paper.

---

> ### Author Rebuttal · Authors · 2026-03-30
>
> # Official Response — Reviewer mBvY
>
> We sincerely appreciate your review and recognition of the performance and the real-world value of our paper. We respond to your questions as follows.
>
> ---
>
> > **W1: Originality of the recurrent memory paradigm**
>
> Thanks for your concern. We would like to highlight that our paper mainly focuses on identifying and solving critical limitations in the existing recurrent memory mechanisms (i.e., memory explosion and lack of exit) and getting promising results, rather than being positioned as a new paradigm.
>
> Additionally, papers that do not propose new paradigms but instead address and solve the limitations of existing work are common.
> For example, Dr.GRPO [1] and DAPO [2] identify limitations of the vanilla GRPO [3] in training by adding new features rather than proposing a new paradigm.
> The GRPO algorithm is not proposed in these papers, but such papers significantly improve the vanilla GRPO by carefully adding new features.
>
> From this perspective, we believe the insights of this paper themselves are non-trivial since they solve the crucial problems, and we are the first work that brings such gating mechanisms into long-context agents.
>
>
> [1] Understanding R1-Zero-Like Training: A Critical Perspective. COLM 2025.
>
> [2] DAPO: An Open-Source LLM Reinforcement Learning System at Scale. NeurIPS 2025.
>
> [3] DeepSeekMath: Pushing the Limits of Mathematical Reasoning in Open Language Models. Arxiv 2024.
>
>
> ---
>
> > **Q1: The necessity of the gating mechanism**
>
> Thanks for your valuable question. We respond to this question as follows. We are also willing to further discuss this question if not fully understand your questions.
>
> **Core weaknesses and limitations of MemAgent & Our solution:** As stated in our introduction, MemAgent has two critical limitations: the risk of memory explosion and the lack of an exit mechanism. The memory explosion phenomenon has also been validated in Figure 6, where the MemAgent quickly reaches the maximum memory size. Therefore, we propose two gating mechanisms to address such issues, where such motivation is also recognized by the other reviewers.
>
> **Why prompt tuning alone cannot fix.** We add two additional experiments to demonstrate why prompt tuning can not fix.
> 1. Running the model trained with MemAgent framework under the GRU-Mem framework (w EG). We report the performance in Table 1.
>
> **Table 1 Running GRU-Mem with the MemAgent model**
> |Method|Metric|HQA|SQuAD|SK-1|SK-2|SK-3|MK-1|MK-2|MK-3|MQ|
> |-|-|-|-|-|-|-|-|-|-|-|
> |GRU-Mem (with MemAgent model)|Perf.%↑|44.83|56.12|99.11|84.26|85.15|77.90|51.68|90.51|46.93|
> ||Time s↓|152.36|40.57|85.20|88.69|89.89|83.86|48.52|73.49|87.73|
>
> As shown, when changing the prompt of the model trained under MemAgent framework, it actually performs badly under the GRU-Mem framework.
> This implies that models trained via MemAgent cannot simply discern from prompts whether the current chunk contains evidence, or whether sufficient evidence has been gathered to generate a response (the reduced inference time indicates an indiscriminative early-stop behavior).
>
> 2. Modifying the prompts of MemAgent and still running under its framework. We add one more line prompt of """Only update your current memory when this chunk contains evidence for answering the question""" (i.e., MemAgent (New Prompt)). We still track the memory dynamics under this setting, with the same setting as in Figure 6.
> We report the memory dynamics as the steps increase in Table 2.
>
> **Table 2 Memory Token Dynamics**
>
> |Method/Step|1|11|21|31|41|51|61|71|81|91|101|
> |-|-|-|-|-|-|-|-|-|-|-|-|
> |MemAgent|158.34|992.09|984.59|924.80|946.04|942.45|960.70|985.23|990.14|999.42|993.32|
> |MemAgent (New Prompt)|154.44|1013.31|1005.77|981.92|996.15|1002.74|996.35|993.29|993.98|995.30|980.45|
> |GRU-Mem|6.55|72.79|88.90|132.76|104.81|114.25|97.79|90.82|93.56|113.55|118.20|
>
> As shown, even when changing the prompts, the vanilla MemAgent still suffers from the memory explosion risks, while GRU-Mem maintains a relatively clean memory.
>
> Therefore, merely tuning the prompt can not effectively address the two main limitations in the MemAgent.
>
> **Why refining the reward under the MemAgent framework cannot fix.** In the vanilla MemAgent, it is hard to design additional rewards for limiting the update behavior. If we add a turn-level reward according to the MemAgent for controlling the update behavior, it just becomes the same as GRU-Mem. Additionally, in the vanilla MemAgent, it is unable to design a reward for controlling the exit behavior. Therefore, it is unable to introduce additional rewards in the vanilla MemAgent to learn such update and exit behaviors.
>
> **Why gates help:** As we have analyzed above, only when introducing the two gating mechanisms and training the model with carefully designed reward signals can we enable the model to update and exit correctly.
>
> We sincerely appreciate your comments, and would like to further clarify if not fully addressing concerns.

---

> > ### Author Rebuttal · Reviewer_mBvY · 2026-04-02
> >
> > I agree authors.

---

> > > ### Author Response · Authors · 2026-04-03
> > >
> > > Thanks for your timely reply! We will refine our latest paper according to your suggestions in the latest version.

---

### Official Review · Reviewer_iJDh · 2026-03-13

**Soundness:** 3
**Presentation:** 3
**Significance:** 3
**Originality:** 3
**Overall Recommendation:** 5
**Confidence:** 4

**Summary:**

This paper presents GRU-Mem, a language-based gated recurrent memory framework that improves long-context reasoning in LLMs. To solve the issues of memory explosion and wasted computation in prior chunk-by-chunk methods, GRU-Mem introduces two mechanisms: an update gate to selectively store relevant information, and an exit gate to stop processing early once sufficient evidence is gathered. Trained via reinforcement learning, GRU-Mem outperforms the MemAgent baseline and accelerates inference by up to 4x.

**Compliance With Llm Reviewing Policy:**

Affirmed.

**Final Justification:**

This paper builds on a nice idea and has clear merits. The only weakness that I can recognize is that it only trains on a single model family (Qwen2.5), and it would be good to verify robustness for additional model families.

**Key Questions For Authors:**

- What is the theoretical or empirical justification for disentangling the advantage calculation into two distinct terms (trajectory-level and turn-level) in Equation 13 instead of computing a single joint advantage?
- In Section 4.2, to provide a more direct analysis of the gate's behavior, could you also plot the average update gate activation frequency across these steps?
- Reinforcement learning appears to provide the most significant performance boosts in standard QA tasks (like HQA and SQuAD) rather than the synthetic MK retrieval tasks (Figure 10). Do you have any hypotheses or insights into why RL is less impactful for multi-key synthetic retrieval?

**Limitations:**

Yes.

**Strengths And Weaknesses:**

## Strengths

- The core idea of implementing a text-based GRU memory is a reasonable, original, and interesting approach to long-context reasoning.
- The two main limitations of MemAgent (memory explosion and lack of early exit) are clearly explained, well-justified, and directly addressed by the proposed gating method.
- The manuscript is clear and well-organized.

## Weaknesses

1. A notable limitation of the current method is that the exit gate needs to be manually disabled (w/o EG mode) for tasks like the multi-values task in RULER where the model needs to process the entire context. This is concerning for the overall generalization and autonomy of the method. Ideally, the reinforcement learning process should teach the model to recognize absolute qualifiers in the prompt (like "all" in "What are all the special magic numbers...") and dynamically learn *not* to early-exit. Relying on a human-in-the-loop to toggle the exit gate diminishes the "agentic" nature of the framework.

2. The current training and evaluation setup relies entirely on the Qwen2.5 model family (3B and 7B). Evaluating the approach on additional, distinct model architectures (e.g., Llama 3 or Phi-3) would greatly strengthen the paper's claims that this method is generally applicable across different foundation models.

---

> ### Author Rebuttal · Authors · 2026-03-30
>
> # Official Response — Reviewer iJDh
>
> We sincerely appreciate your positive assessment of our motivation, clarity, and empirical results.
> We also thank you for raising important suggestions and concerns. We respond to your questions as follows.
>
> ---
>
> > **W1: How to choose the correct inference mode autonomously**
>
> Thanks for the constructive suggestions.
>
> Actually, this can be easily solved by adding one additional LLM-based judge before solving the question with the following prompt (some details are omitted due to word limits):
>
> """... Determine whether this question allows early stopping.
> Answer Yes only if the answer can already be known to be complete without reading the rest of the document.
> Answer No if ... require reading the whole document to be certain.
> Return only one word: Yes or No.
>
> Question:
> {prompt}"""
>
> We activate the EG mode if receiving Yes, and deactivate it otherwise.
> We report the accuracy of choosing the correct inference mode and corresponding performance in Table 3. We primarily ignore such engineering optimization in our paper to stay focused on the algorithm improvement.
>
> **Table 3: Inference Mode Selection**
> |Method|Metric|HQA|SQuAD|SK-1|SK-2|SK-3|MK-1|MK-2|MK-3|MQ|MV|
> |-|-|-|-|-|-|-|-|-|-|-|-|
> |GRU-Mem|Mode Acc.%↑|97.66|92.19|100.0|100.0|100.0|100.0|100.0|100.0|100.0|100.0|
> ||Perf.%↑|76.56|82.29|100.00|94.64|96.54|98.10|82.48|95.65|85.16|95.34|
> ||Time s↓|252.53|79.34|149.28|140.97|135.73|129.19|148.93|148.47|134.88|178.99|
>
> As shown, the model itself can correctly choose the right inference mode (i.e., w/o EG for MV and EG for others). Additionally, it is possible that stronger models would be more capable of identifying the correct inference mode.
>
>
> ---
>
> > **W2: Training and evaluation primarily on Qwen2.5 (3B and 7B)**
>
> Thanks for your concerns. It is a good suggestion to extend our method to more backbones.
> Currently, we primarily conduct experiments on Qwen models due to the following practical concerns.
>
> 1. Qwen is the most stable model among open-sourced ones for RL training. Prior literature has found that Llama and other models may not be stable for RL training [1,2]. To enable effective RL training, we commonly need to conduct continued finetuning on such models as Llama before RL. As a result, verifying our method on these models requires much more probing and computational costs, which exceed our current computational resource capacity.
> 2. Training on the Qwen 2.5 series aligns well with the original MemAgent paper, which provides more fair comparison.
>
> Therefore, we mainly focus on the Qwen 2.5 series. We leave exploration of other models to future work.
>
> [1] OctoThinker: Mid-training Incentivizes Reinforcement Learning Scaling. Arxiv 2025.6.
>
> [2] Understanding R1-Zero-Like Training: A Critical Perspective. COLM 2025.
>
> ---
>
> > **Q1: Justification of advantage disentanglement**
>
> Thank you for the valuable question.
> The advantage disentanglement makes the training more flexible and stable, as we can use a hyperparameter alpha to control the effect of each part.
> The ablation study in Figure 9 also shows that the hyperparameter alpha plays a crucial role.
> If combining the advantages, we cannot manually balance the update gate and exit gate.
> As a result, this may lead to sub-optimal performance.
>
> More discussion about similar advantage disentanglement can be found in GDPO [1].
>
> [1] GDPO: Group reward-Decoupled Normalization Policy Optimization for Multi-reward RL Optimization. Arxiv 2026.1
>
> ---
>
> > **Q2: Average update gate activation frequency across steps**
>
> Thanks for your suggestions.
> We add the update gate frequency of the 3B-size model in Table 2, since the update frequency is not tracked during our 7B-size model training.
> We would also further add more figures to our latest paper.
>
> As shown, the update gate frequency is generally around 0.35, demonstrating that GRU-Mem selectively activates the update gate.
>
> |Step|10|20|30|40|50|100|150|200|250|300|350|400|450|500|
> |-|-|-|-|-|-|-|-|-|-|-|-|-|-|-|
> |Update Rate|0.001200|0.039594|0.077624|0.241434|0.290530|0.257803|0.261230|0.278623|0.326295|0.328337|0.333333|0.321786|0.300779|0.342484|
>
> ---
>
> > **Q3: RL appears to help more on standard QA (HQA, SQuAD) than on synthetic MK retrieval (Figure 10)**
>
> Thanks for the insightful observation. We attribute this to two factors. On the one hand, the QA task shows relatively low performance before training, resulting in more significant improvements. On the other hand, QA tasks are relatively in-distribution, whereas MK-series tasks are out-of-distribution.

---

> > ### Author Rebuttal · Reviewer_iJDh · 2026-04-04
> >
> > Thanks for the new results and clarifications. Regarding the models, I still believe that models like Phi4 and Qwen3 should be fairly easy to fine-tune via RL (and also at small sizes). However, I think the results are interesting and strong enough even without testing on a different model family (hence my initial score). I keep my current score.

---

> > > ### Author Response · Authors · 2026-04-07
> > >
> > > Thanks so much for your positive feedback! We will add your suggestions to our latest paper.

---

### Official Review · Reviewer_NwGq · 2026-03-13

**Soundness:** 2
**Presentation:** 3
**Significance:** 2
**Originality:** 3
**Overall Recommendation:** 4
**Confidence:** 4

**Summary:**

This paper investigates the recurrent memory mechanism in long-context reasoning. It points out that while the existing method, MemAgent, handles ultra-long contexts by reading in chunks and progressively updating memory, it suffers from two issues: it continuously updates on irrelevant chunks, leading to memory bloat, and it lacks an early stopping mechanism, resulting in unnecessary computational overhead. To address these issues, the paper proposes GRU-Mem, which introduces an update gate and an exit gate at each step to respectively control whether to update the memory and whether to terminate the scanning early once sufficient evidence is gathered.

In terms of training, in addition to the final answer reward, the authors designed specific rewards for the updating and exiting behaviors, optimizing the entire pipeline through end-to-end reinforcement learning. Experimental results demonstrate that GRU-Mem generally outperforms the vanilla MemAgent across various long-context reasoning tasks while significantly enhancing inference efficiency, achieving approximately a 4x speedup under certain settings. Overall, the primary contribution of this paper is explicitly modeling "when to remember and when to stop" as critical decisions in long-context reasoning, and validating the benefits of this gated design for both performance and efficiency.

**Compliance With Llm Reviewing Policy:**

Affirmed.

**Final Justification:**

My doubts were resolved and I recognized the value of the article.

**Key Questions For Authors:**

1. On Baselines and Comparative Scope. The paper currently primarily compares the proposed method with the vanilla MemAgent. Could the authors further clarify why a broader or stronger set of long-context baselines (such as representative long-context prompting, retrieval-based, or other memory-based methods) was not included? If additional experiments cannot be provided at this time, I would appreciate an explanation regarding the considerations for fairness and persuasiveness within this current comparative scope.

2. On Reward Design and Signal Acquisition. The update reward in the paper requires determining whether a specific chunk contains evidence relevant to the question, while the exit reward requires knowing the position of the "last necessary piece of evidence." Could the authors provide more detail on how these evidence-level signals are obtained during training? For instance, do they come from manual annotations, dataset-provided supporting facts, string-matching rules, or other heuristic methods? Furthermore, is this annotation or construction method consistent across different benchmarks, and might it limit the scalability of the method to real-world, open-ended scenarios?

3. On the Application of the Exit Gate. The paper provides two inference modes: with exit gate and without exit gate, as some tasks require reading the entire context. Could the authors further explain how to determine whether it is appropriate to enable the exit gate during testing, particularly in practical use cases where the task requirements may not be fully explicit beforehand?

**Limitations:**

Not fully. The authors do discuss some limitations, including the focus on QA-style tasks and the reduced training stability caused by additional rewards, and they briefly mention the risk of hallucinations. However, the discussion could be more complete. In particular, the paper should more clearly acknowledge its dependence on evidence-level supervision for reward construction, the risk of premature stopping when full-context aggregation is required, and the potential consequences of incorrect memory updates or exit decisions in real-world settings. Overall, the limitations section is helpful but still somewhat limited in scope.

**Strengths And Weaknesses:**

Strengths: The paper successfully identifies two critical issues in long-context recurrent memory: memory bloat caused by indiscriminate updates and computational waste due to the lack of an early exit mechanism. The proposed dual-gate design is both intuitive and reasonable, maintaining strong self-consistency when paired with the reinforcement learning rewards. The experiments—covering performance, efficiency, memory length, and exit behavior—effectively support the main conclusions. In terms of significance, this work offers practical value for scenarios like long-document Q&A and agent memory, while providing a promising research direction in modeling "when to remember and when to stop."

Weaknesses: The methodological novelty is moderate; it feels more like a natural enhancement of the existing MemAgent framework rather than a brand-new paradigm. The experiments are primarily focused on Q&A-style tasks, leaving its generalizability somewhat limited. Furthermore, there is room for improvement in the presentation: the main figure does not clearly illustrate how the gates control information flow and the exit process, and some implementation details necessary for reproducibility are insufficient.

---

> ### Author Rebuttal · Authors · 2026-03-30
>
> # Official Response — Reviewer NwGq
>
> We sincerely appreciate your thoughtful review and respond to your comments as follows.
>
> ---
>
> > **W1: Novelty**
>
> We would like to clarify that our core contribution is not positioned as a new paradigm, but rather as identifying and solving critical limitations in existing recurrent memory mechanisms (i.e., the memory explosion and lack of exit) and achieving promising results.
> We believe such insights themselves are non-trivial as they address crucial problems, and we are the first work that brings such gating mechanisms into long-context agents.
> Additionally, papers that do not propose new paradigms but instead identify and address the limitations of existing work are common.
> For example, Dr.GRPO and DAPO identify limitations of the vanilla GRPO in training by introducing targeted improvements rather than proposing a new paradigm.
>
> ---
>
> > **W2: Q&A tasks**
>
> We focus on QA tasks for two reasons:
>
> 1. QA is the most common long-context reasoning scenario, as other tasks like summarization rarely involve contexts reaching 1M.
> 2. RL training requires task-specific reward design and data, so we limit our scope to QA and leave extension to other tasks for future work.
>
> To address your concern, we added a summarization experiment on QMSum with our 3B model. As shown in Table 1, our method performs comparably to MemAgent and outperforms the vanilla LLM, despite not being trained on summary tasks.
>
> **Table 1: Performance on QMSum**
> ||ROUGE-1|ROUGE-2|ROUGE-L|AVG|
> |-|-|-|-|-|
> |LLM|0.2760|0.0672|0.1820|0.1751|
> |MemAgent|0.3221|0.0954|0.2096|0.2091|
> |GRU-Mem|0.3313|0.0859|0.2138|0.2103|
>
>
> ---
>
> > **W3: Presentation clarity**
>
> We have revised the main figure in our latest paper.
> Additionally, we summarize how our method controls the information flow in one sentence: at each step, the memory agent decides whether to update the memory with the Ut signal, and whether to exit the loop with the Et signal.
>
> ---
>
> > **Q1: Baselines comparison**
>
> We mainly compare our GRU-Mem with MemAgent for the following reasons:
>
> 1. In the MemAgent paper, it was demonstrated that the vanilla MemAgent can outperform long-context prompting and RAG methods.
> 2. Our method mainly focuses on addressing the two limitations in the RNN-like MemAgent, so our experiments are designed to validate our claims.
>
> Therefore, we mainly compare with the MemAgent method. To further address your concerns, we add one more representative workflow-based baseline Mem0 [1] in Table 2.
>
> **Table 2: Performance Comparison**
> |Method|Metric|HQA|SQuAD|SK-1|SK-2|SK-3|MK-1|MK-2|MK-3|MQ|
> |-|-|-|-|-|-|-|-|-|-|-|
> |Mem0|Perf.%↑|73.15|70.38|100.00|94.87|94.87|96.88|51.56|58.82|78.10|
> ||Time s↓|794.00|281.52|458.44|381.56|392.80|352.76|291.44|336.48|400.16|
> |GRU-Mem|Perf.%↑|76.37|80.47|100.00|96.65|95.20|98.55|84.15|95.54|84.12|
> ||Time s↓|209.33|64.32|126.00|113.89|107.64|102.46|123.53|136.02|108.62|
>
> As shown in this table, our method outperforms Mem0 while being much faster. We attribute this to Mem0's overly complex workflow and its lack of specialized training.
>
> [1] Mem0: Building Production-Ready AI Agents with Scalable Long-Term Memory. Arxiv 2025.4
>
> ---
>
> > **Q2: Reward design and signal acquisition**
>
> Thanks for the valuable questions. Actually, these evidence-level signals are straightforward to obtain.
> The original dataset already contains the evidence required to answer each question.
> Accordingly, we only need to conduct string matching to identify the specific chunks across which the evidence is distributed.
>
> Thus, our method requires no human annotation beyond the original dataset.
> Since almost all existing QA datasets follow this organizational form, our approach applies to most real-world scenarios.
>
> ---
>
> > **Q3: How to choose the correct inference mode autonomously**
>
> Thanks for the constructive suggestions.
> Actually, this can be easily solved by adding one additional LLM-based judge before solving the question with the following prompt (some details are omitted due to word limits):
>
> """... Determine whether this question allows early stopping.
> Answer Yes only if the answer can already be known to be complete without reading the rest of the document.
> Answer No if ... require reading the whole document to be certain.
> Return only one word: Yes or No.
>
> Question:
> {prompt}"""
>
> We activate the EG mode if receiving Yes, and deactivate it otherwise.
> We report the accuracy of inference mode selection and the corresponding performance in Table 3.
>
> **Table 3: Inference Mode Selection**
> |Method|Metric|HQA|SQuAD|SK-1|SK-2|SK-3|MK-1|MK-2|MK-3|MQ|MV|
> |-|-|-|-|-|-|-|-|-|-|-|-|
> |GRU-Mem|Mode Acc.%↑|97.66|92.19|100.0|100.0|100.0|100.0|100.0|100.0|100.0|100.0|
> ||Perf.%↑|76.56|82.29|100.00|94.64|96.54|98.10|82.48|95.65|85.16|95.34|
> ||Time s↓|252.53|79.34|149.28|140.97|135.73|129.19|148.93|148.47|134.88|178.99|
>
> As shown, the model itself can correctly choose the right inference mode (i.e., w/o EG for MV and EG for others).

---

> > ### Author Rebuttal · Reviewer_NwGq · 2026-04-03
> >
> > I will raise my score to 4 points.

---

> > > ### Author Response · Authors · 2026-04-03
> > >
> > > Thank you for your recognition! We will add your suggestions in our latest paper.

---

### Decision · Program_Chairs · 2026-04-30

**Decision:**

Accept (regular)

**Comment:**

This paper proposes GRU-Mem, a gated extension of MemAgent for long-context reasoning that introduces an update gate to control memory growth and an exit gate to stop processing once sufficient evidence has been collected. Reviewers generally found the problem important, the design intuitive and well-motivated, and the empirical results strong, particularly the substantial efficiency gains and improved memory stability. The main concerns were that the methodological novelty is moderate because the approach is largely an incremental extension of MemAgent rather than a fundamentally new paradigm, the evaluation is concentrated mostly on Q&A-style long-context tasks, and generalization to broader settings remains unclear. Reviewers also noted that the exit gate sometimes needs to be manually disabled for tasks requiring full-context processing, which raises questions about robustness and autonomy, and that experiments are limited to the Qwen2.5 family, so validation on additional model families would strengthen the claims. Some concerns were also raised about presentation and reproducibility, especially around explaining the gate behavior and providing sufficient implementation detail. Overall, the paper was viewed positively as a practically useful and well-executed improvement over prior recurrent-memory approaches, with the main reservation being limited novelty.